# Optical and Transport Properties of ZnO Thin Films Prepared by Reactive Pulsed Mid-Frequency Sputtering Combined with RF ECWR Plasma

**DOI:** 10.3390/nano15080590

**Published:** 2025-04-11

**Authors:** Zdeněk Remeš, Zdeněk Hubička, Pavel Hubík

**Affiliations:** FZU—Institute of Physics of the Czech Academy of Sciences, Na Slovance 1999/2, 182 00 Prague, Czech Republic; hubicka@fzu.cz (Z.H.); hubik@fzu.cz (P.H.)

**Keywords:** ZnO, optical absorptance, photoluminescence, sputtering, Hall effect

## Abstract

The study explores the optical and transport properties of polycrystalline ZnO thin films prepared using reactive pulsed mid-frequency sputtering with RF electron cyclotron wave resonance (ECWR) plasma. This deposition method increases the ionization degree of sputtered particles, the dissociation of reactive gas and the plasma density of pulsed reactive magnetron plasma. Optical absorption spectra reveal a sharp Urbach edge, indicating low valence band disorder. Lattice disorder and deep defect concentration are more likely to occur in samples with higher roughness. PL analysis at low temperature reveals in all samples a relatively slow (μs) red emission band related to deep bulk defects. The fast (sub-ns), surface-related blue PL band was observed in some samples. Blue PL disappeared after annealing in air at 500 °C. Room temperature Hall effect measurements confirm n-type conductivity, though with relatively low mobility, suggesting defect-related scattering. Persistent photoconductivity was observed under UV illumination, indicating deep trap states affecting charge transport. These results highlight the impact of deposition and post-treatment on polycrystalline ZnO thin films, offering insights into optimizing their performance for optoelectronic applications, such as UV detectors and transparent conductive oxides.

## 1. Introduction

ZnO has emerged as a promising, eco-friendly material that offers unique optical properties and a range of nano and microstructures that can be applied in fields such as photovoltaic energy conversion, photocatalytic wastewater treatment, electrochemical energy storage, thin-film optoelectronics, scintillators, piezoelectronic devices, transparent electronics, spintronics, and sensors [1,2]. ZnO is a direct wide band gap semiconductor, transparent for visible light and suitable for UV optoelectronic applications [3]. The Zn-O bonds are sp^3^ hybridized of almost equally ionic and covalent character [4]. Zinc oxide crystallizes in the hexagonal wurtzite structure (space group P6_3_mc) with the lattice parameters *a* and *c* equal to 325 pm and 521 pm, respectively [5], where each Zn^2+^ ion is tetrahedrally coordinated by four O^2−^ ions, and vice versa. This structure is characterized by a non-centrosymmetric lattice, which gives rise to ZnO’s inherent piezoelectric and pyroelectric behavior due to its polar nature along the c-axis [6]. ZnO exhibits strong near-edge excitonic photoluminescence. The UV emission around 3.37 eV at room temperature is mainly attributed to free excitons or excitons bound to neutral donors, reflecting the high exciton binding energy (~60 meV) inherent to the wurtzite ZnO crystal structure [7]. In addition to the near-band-edge ultraviolet emission, ZnO often exhibits broad visible photoluminescence generally attributed to intrinsic point defects, such as zinc interstitials (Zn_i_) or zinc vacancies (V_Zn_), which act as recombination centers for electrons and holes. The green luminescence centered between 2.5 and 2.6 eV was often observed in ZnO thin films [8,9]. It was suggested that it may originate from transitions between electrons in the conduction band and zinc vacancy levels [10,11], recombination of a shallowly trapped electron with a hole in a deep trap [12] or the recombination of an electron in singly occupied oxygen vacancies with a photo-generated hole in the valence band [13]. A transition from an intrinsic shallow state to an intrinsic deep state was suggested as an origin of red PL in ZnO thin films [14]. Newer studies suggest that defect-related PL in ZnO likely originates from defect complexes rather than single point defects [15]. It was suggested that oxygen vacancies and interstitials may contribute to the intrinsic n-type conductivity of ZnO by providing free electrons [16]. However, recent theoretical studies suggest that isolated oxygen vacancies may not be effective shallow donors and instead act as deep-level traps [17,18]. The n-type conductivity in doped ZnO films is mainly due to group III impurities (In, Ga, B, Al) substituting Zn^2+^ [19,20]. The typical values from 10^16^ to 10^18^ cm^−3^ of free electron concentration and 100–200 cm^2^V^−1^s^−1^ of electron mobility were observed in dark at room temperature in high-quality undoped (intrinsic) crystalline ZnO [21,22]. Hydrogen acts as a shallow donor with 35 meV ionization energy at the interstitial position incorporating in ZnO [23,24]. The exciton-related photoluminescence in near UV region was significantly enhanced by inductively coupled hydrogen plasma at room temperature, whereas the broad-band defect related photoluminescence was reduced [25].

In our previous research, we studied several micrometers-thick, undoped ZnO thin films deposited onto glass substrates without a buffer layer [26]. This deposition was achieved using an innovative hybrid pulsed reactive magnetron sputtering system operating in the medium-frequency (MF) range, enhanced with electron cyclotron wave resonance (ECWR) plasma assistance. As already presented in [26,27,28], the advantage of an additional RF ECWR plasma source lies in a higher degree of reactive gas (oxygen) dissociation and consequently higher reactivity of atomic oxygen with Zn atoms deposited on the substrate. It has also been shown that the higher plasma density at higher ECWR absorbed discharge power can influence the crystallization of deposited films [26,28] at lower substrate temperature. Another advantage of ECWR plasma is that it allows stable pulsed magnetron discharge at lower pressure than conventional pulsed magnetron. Lower pressure of argon gas during deposition results in higher energy of sputtered particles, and in certain cases can produce denser structure with lower defect concentration. The resulting ZnO films exhibited remarkably fast UV photoluminescence (PL) decay times and a significant reduction in defect-related visible luminescence at room temperature, marking the fastest decay times ever reported for this system. These impressive characteristics were attributed to the advanced deposition method, which allows for precise control over plasma parameters. Specifically, the high level of molecular oxygen dissociation and the enhanced activation of zinc species contribute to faster deposition rates, better crystalline structure, fewer structural defects, and a reduced concentration of oxygen vacancies, all of which result in superior optical performance.

The present work deals with further studies of ZnO thin films deposited by MF pulsed reactive magnetron sputtering under similar conditions as presented in [26,27,28]. Other properties such as Hall mobility, carrier concentration, and absorption coefficient in the low absorption region have been measured and discussed in relation to the deposition parameters. Bulk and surface PL were distinguished by excitation wavelengths of 340 nm (top) and 385 nm (below the optical absorption edge).

## 2. Materials and Methods

### 2.1. Hybrid Mid-Frequency Sputtering Assisted by an Electron Cyclotron Wave Resonance Plasma (MFS and ECWR)

Thin film ZnO depositions on soda lime glass substrates (samples #1–#5) and fused silica glass substrates (samples #6, #7) were performed in a high vacuum (HV) stainless steel chamber using hybrid mid-frequency sputtering supported by a high frequency electron cyclotron wave resonance plasma (MFS and ECWR) deposition system [26,27,29]. The deposition system was equipped with a 2-inch magnetron with a 50 mm diameter circular planar cathode. The 6 mm thick target was made of 99.9% pure zinc (Plasmaterials, Inc., Livermore, CA, USA). The magnetron was connected to a bipolar pulsed power supply with a pulse frequency of 40 kHz. A 200 mm diameter water-cooled radio frequency (13.56 MHz) inductively coupled plasma (RF ICP) coil was placed between the magnetrons and the substrates. The total absorbed MFS average power in the pulsed magnetron discharge was 170 W for samples #1–#6 and 280 W for sample #7. The RF power applied to the ECWR coil is shown in Table 1 and was set in the range of 0–380 W for samples #1–#5 deposited on soda lime glass. The samples #6 and #7 were performed on fused silica for specific ECWR power 200 W, as this value was shown to be best for the photoluminescence study conducted in [26]. A static homogeneous magnetic field with a magnetic induction of ≈1.7 mT was introduced into the ECWR electrode region using a pair of coils to bring the system close to ECWR. For all samples presented in Table 1, the argon gas flow was 10 sccm, and the oxygen gas flow was 10 sccm, and the working gas pressure was 0.3 Pa. The distance between the magnetron target surface and the substrate was 110 mm. The substrates were heated to 350 °C by an external heater during deposition.

### 2.2. Optical Transmittance, Reflectance, and Photothermal Deflection Spectroscopy (PDS)

The transmittance, reflectance, and absorbance spectra were measured simultaneously in the 300–1400 nm spectral range using a photothermal deflection spectroscopy (PDS) setup [30,31]. During the optical measurements, the samples were immersed in liquid (Florinert FC72, 3M Company, St. Paul, MN, USA) to measure the relative temperature of the illuminated sample independently for selected photon energies by deflection of a probe laser beam. Quazi monochromatic light was provided by a 150 W Xe lamp and a monochromator (SpectraPro-150, Acton Research Corp., Acton, MA, USA) equipped with two gratings: a UV holographic grating (1200/mm) and a ruled grating (600/mm) blazed at 500 nm and with slits of 1/1 mm. The spectra were spectrally calibrated by measuring the PDS of a black carbon sample. The spectral resolution was 5 nm for the UV holographic grating and 10 nm for the ruled grating.

### 2.3. Steady-State Photoluminecence at Low Temperature (Steady-State PL)

The steady state PL spectra were measured at low temperature with samples mounted directly on a copper holder in a Cryofree^®^ optical cryostat OptistatDry BLV (Oxford Instruments, Tubney Woods, Abingdon, Oxon, OX13 5QX, England) providing a temperature-controlled, sample-in-vacuum measurement environment at low temperature (nominally 4 K—the Cu stage temperature, the sample may have higher temperature due to the heating by excitation light). PL was excited by fiber coupled UV LEDs M340F3 and M385F1 (Thorlabs, Inc., Newton, NJ, USA) at wavelengths of 340 nm (1 mW) and 385 nm (10 mW) at a frequency 307 Hz and illuminated spot size diameter about 0.5 mm. Both LEDs were equipped with 10 nm FWHM (Full Width at Half Maximum) band pass filters (Edmund Optics, Inc., Barrington, IL, USA). The emission spectra were measured in the spectral range 360–780 nm (excitation LED340) or 400–780 nm (LED385) using the perpendicular geometry with a long pass optical filters LP350 or LP400 (Edmund Optics, Inc., Barrington, IL, USA), placed in front of a spectrally calibrated double gratings monochromator SPEX1672 (SPEX Industries, Inc., Edison, NJ, USA). PL was detected by multi-alkali photomultiplier tube (PMT) XP2203B (Photonis, Mérignac, France), 10^5^ V/A transimpedance low noise current preamplifier #5184 (AMETEK, Berwyn, PA, USA) and a lock-in amplifier #5105 (AMETEK, Berwyn, PA, USA) referenced to the LED frequency. The PMT was cooled to −12 °C to decrease the dark dc current by one order of magnitude from about 2 nA at room temperature. The ac photocurrent noise of cooled PMT at 307 Hz and −1600 V bias was about 10 pA corresponding to 1 µV noise at lock-in voltage input. The PL setup was spectrally calibrated with a #63358 halogen lamp (Oriel Instruments, subsidiary of Newport Corporation, Stratford, CT, USA). The additional correction [32] of the steady-state setup was related to the Jacobian conversion of wavelength and energy scales for quantitative analysis of emission spectra.

### 2.4. Frequency-Resolved Photoluminecence at Room Temperature (f-Resolved PL)

The frequency resolved PL was measured only at room temperature using a pulsed UV LED XSL-360-5E (Roither Lasertechnik GmbH, Vienna, Austria) at a variable frequency up to 1 MHz. The UV LED was equipped with UV band pass filter BP360(10) (transparent in 350–370 nm spectra range) directly powered by signal output (sinusoidal amplitude 1.2 V, DC offset 4.5 V) of a lock-in amplifier HF2LI (Zurich Instruments AG, Zurich, Switzerland). The PL emission was measured in the 375–760 nm spectral range using the long-pass optical filters LP375 and LP500 fully absorbing below 370 nm (500 nm), monochromator H20VIS (F/4.2, 1200 gr/mm, linear dispersion 4 nm/mm), red-sensitive Philips Photonics XP2203B photomultiplier (PMT) and HF2LI lock-in amplifier referenced to the LED frequency and equipped with transimpedance preamplifier HF2TA. The dark dc current of PMT (bias −1000 V) was about 0.5 nA. The setup was suitable for time resolved PL measurements with the time resolution above 1 ns using the method based of the phase shift between the sinusoidal excitation and emission [33]. Prior to the phase shift measurements, the lock-in amplifier was set to zero phase at the excitation wavelength of 360 nm. The setup was spectrally calibrated with ORIEL QTH #63358 halogen lamp (Oriel Instruments, subsidiary of Newport Corporation, Stratford, CT, USA).

### 2.5. Hall and Photo-Hall Measurements

Resistivity, Hall, and photo-Hall measurements were obtained using the van der Pauw method. The ZnO films were provided with four indium soldered contacts in the corners of rectangular (cca 10 mm × 12 mm) samples. The linearity and symmetry of the contacts were verified. Samples were placed in a metallic chamber on a PTFE support and connected to the measuring circuit by phosphor bronze springs. The chamber has featured a quartz window with a removable cap and internal mirror directing the illumination passing through the window to the sample surface. In the measurement, a Keithley 6430 source-meter (Keithley Instruments, LLC, Solon, OH, USA) was used as a current source, two Keithley 6514 electrometers read potential values at voltage contacts, and a Keithley 2182A nanovoltmeter measured the difference in electrometers’ analog outputs to obtain resulting voltage. Triax cables and connectors were used to suppress parasitic cable charging and current leakages. All measurements were performed at room temperature.

For Hall and photo-Hall experiments, we used magnetic field of ±0.2 T, generated by GMW 5403 electromagnet (GMW Associates, San Carlos, CA, USA) powered by a dual Kepco BOP 20-20M power supply (Korea Electric Power Corporation, Naju, South Jeolla Province, Republic of Korea). To measure photo-resistivity and photo-Hall effect, we applied illumination from M365F1 LED by Thorlabs generating light with peak intensity at 365 nm, using the LED’s maximum output power of 5 mW. The light beam covered approximately the full sample area.

## 3. Results

### 3.1. Optical Transmittance, Reflectance and Absorption Spectra

The optical transmittance (*T*), reflectance (*R*), and absorptance (*A*) of two selected ZnO thin films are shown in Figure 1. For comparison, the spectrum 1-*T*-*R*, heavily influenced by the optical scattering related to grain boundaries and surface roughness, was also added. The film thickness was evaluated from the interference fringes in *R* spectra in the region of low optical absorption below 2.5 eV, while fitting the dielectric function ε(*E*) by Lorenc formula for damped harmonic oscillator, see Equation (1), where *E*_0_ is the central position of the Lorenc peak, *E*_1_ the damping energy, ε_∞_ the low energy limit of ε(*E*) and *A* the amplitude [34,35]. The root-mean-square (rms) of surface roughness was evaluated from the suppression of the interference fringes in the reflectance spectra in the near-infrared region using the effective media approximation (EMA) model [36]. It was empirically shown that for the rough surfaces with the high spatial frequencies the *rms* is approximately halve of EMA thickness [37], which approximately coincides with the result achieved theoretically for the one-dimensional rough surface at normal incidence of light in Ref. [38]. Once the real part of the complex index of refraction *n* + i*k* and the film thickness *d* was known, the optical absorption coefficient α(E) was evaluated from PDS spectra independently at each measured photon energy *E*, see Figure 2.(1)$$\varepsilon \left(E\right)={\left(n(E)+ik(E)\right)}^{2}=\varepsilon_{\infty }\left(1+\frac{A^{2}}{E_{0}^{2}-E^{2}+iEE_{1}}\right)$$

Figure 2 shows the optical absorption coefficient spectra of the ZnO layers listed in Table 1. Unlike optical absorptance measured as 1-T-R using transmittance T and reflectance R spectra, PDS measures temperature on the sample surface due to optical absorption. It is therefore much less sensitive to optical scattering and the optical absorption is detectable down to 10^−4^ corresponding to the optical absorption coefficient 10 cm^−1^ for about 1 μm thick film on glass substrate. The optical absorption coefficient was evaluated from PDS spectra using the previously calculated index of refraction and thin film thickness. Th error bar of the absolute value of the optical absorption coefficient shown in Figure 2 was estimated from the residual interference fringes observable in the optical absorption coefficient spectra. The optical absorption coefficient spectra shown in Figure 2 between 1000 and 10,000 cm^−1^ were used to evaluate the exponential Urbach edge [39].

### 3.2. Steady State Photoluminescence Spectra at Low Temperature

The PL spectra of the ZnO layers #1–#5 measured at low temperature are shown in Figure 3. It should be noted that only sample #2 showed the measurable room temperature PL that was about two orders of magnitude weaker than PL at low temperature. No green PL was observed in any sample. Instead, the PL spectrum was dominated by red and blue bands, giving the PL a magenta appearance. All samples show similar broad band red PL band with the maximum at about 1.8 eV (690 nm). In our previous study we have shown that annealing in air at 350 °C resulted in the suppression of red PL, whereas the exciton band became improved. Further increase in the annealing temperature to 700 °C had opposite effect—the exciton band was strongly suppressed whereas the red band intensity increased and was redshifted from about 620 to 680 nm. Red bands we attributed in our previous paper to neutrally charged zinc vacancies (V_Zn_^0^) unperturbed and perturbed by some defect nearby [40].

The steady-state PL spectra measured at low temperature under the excitation wavelength 340 nm and 385 nm before (a) and after (b) 15 min annealing in air at 500 °C of two selected thin films are shown in Figure 4 and in Figure 5. The UV light with the wavelength of 340 nm is heavily absorbed at the surface exciting predominantly only surface-related PL. The blue PL observed in sample 6 diminishes after annealing in air, whereas the red PL is similar in both samples before and after annealing. The illumination with the wavelength of 385 nm is absorbed homogeneously in a sample being therefore related more to bulk properties. The blue PL is negligible under 385 nm excitation and the red PL increases significantly after annealing in air.

### 3.3. Frequency Resolved Photoluminescence Spectra

The frequency spectra of sample #2 measured at room temperature at selected wavelengths show the fast PL in the near UV and blue regions. In our previous paper [41], we discussed that in the case of the mono-exponential decay $PL\left(t\right)\sim e^{-t/\tau }$, the mean time τ of PL decay can be expressed by the formula [33](2)$$\tau =\frac{\tan∆\varphi }{2\pi f}$$
where ∆φ is the phase delay between the excitation and emission at the frequency *f* (in Hz). Since the phase resolution in our setup was about 0.2°, Equation (2) implies the time resolution of about 1 ns at 1 MHz frequency. Figure 6 shows that there is no detectable decrease in the relative amplitude and no detectable phase shift between UV excitation and UV or violet emission up to 1 MHz. The excitation and emission phase shift 10° (30°) at 10 kHz (1 MHz) implies the meantime decay of red PL emission to be about 3 µs (0.1 µs). Thus, the PL decay is not mono exponential and Equation (2) gives only a mean time of PL decay which depends on frequency and the frequency resolved phase shift between the excitation and emission needs to be measured [42].

### 3.4. Hall and Photo-Hall Effect Measurements

A stable Hall signal was obtained in dark at room temperature only after several hours from inserting the sample into the measurement chamber. The thickest sample #7 showed free electron concentration about 10^17^ cm^−3^ and resistivity 1.1 × 10^2^ Ω.cm. The electron mobility was calculated to be about 1 cm^2^V^−1^s^−1^. Thus, sample #7 is n-type with a relatively low carrier concentration and poor electron mobility. Sample #6 was too resistive (4.6 × 10^6^ Ω.cm) in the dark to be able to measure the Hall effect.

Under 5 mW.cm^−2^ UV illumination (LED365), the electrical resistivity decreased by about two orders of magnitude. However, the photocurrent was unstable, and therefore it was not possible to do reliable Hall effect measurements. The long-term persistent photoconductivity suggest the presence of high concentration of electron traps within the band gap [43]. We observed that after switching on the illumination, photoconductance reached a maximum, and then decreased slightly, while it did not reach a completely stable value even within several tens of minutes. This caused the photo-Hall signal, measured against the background of drifting photocurrent, to be rather noisy. This circumstance, together with the not fully optimized homogeneity of the light beam, prevented us from performing extensive photo-Hall measurement, as it would not be sufficiently reliable for the detailed analysis of the persistent conductivity phenomenon. We plan further measurements devoted to the time dependence of conductivity and Hall effect in ZnO prepared by the described method using different illumination and other environmental conditions, which could help us to improve the measurement settings for detailed investigation of the photo-Hall effect in low-mobility samples.

## 4. Discussion

It can be seen in Table 1 that the deposition rate fluctuates with the ECWR power. We assume that at higher power, the Zn target surface has been heated more and evaporation starts to play a role, as already observed in [26]. On the other hand, as the Zn target is sputtered and the Zn target changes its shape in the racetrack, the surface oxidation may fluctuate with different PECWR values and target shape. This different target surface oxidation can affect Zn evaporation and cause this deposition rate fluctuation.

The high correlation coefficient *R* = 0.93 between the Urbach edge slope E_U_ and the optical absorption coefficient α(2.5 eV) indicates the high degree of correlation between valence band disorder and the concentration of deep defect states [44]. The significant correlation has been found also between sample thickness and rms roughness (*R* = 0.76), α(2.5 eV) and rms roughness (*R* = 0.74), between E_U_ and rms roughness (*R* = 0.73) and between E_U_ sample thickness and (*R* = 0.74). The Urbach energy extracted from the absorption spectra increases with increasing lattice disorder, indicating stronger tailing of the density of states into the band gap. The optical absorption coefficient at energy 2.5 eV, well below the optical absorption edge at 3.3 eV, is related to deep electron states in the energy gap acting as recombination centers [45]. It appears that lattice disorder and deep defect concentration are more likely to occur in samples with higher surface roughness.

Defects in ZnO include oxygen vacancies, zinc vacancies, interstitials, and antisite defects [46]. These defects create localized states within the band gap, which can trap charge carriers and affect the material’s conductivity and luminescence properties [45]. There was an exciton-related optical absorption peak at 3.3 eV observed in the sample #6 [47]. In other samples the excitons were not detected in optical absorption spectra. It should be noted that the thinner samples with thickness of about 100 nm would be more appropriate for PDS measurements near the optical absorption edge of ZnO thin films because it is rather difficult to evaluate the optical absorption coefficient from PDS spectra in the region of strong absorption where PDS signal saturates. We never observed green PL in our samples. Instead, there were two dominant PL bands observed in our samples under UV excitation: the blue and red PL. The red PL with a relatively slow mean decay time of several μs was observed in all samples. It is related to bulk properties. The red PL increases after annealing in air. In our previous paper we related the red PL to the oxygen vacancies next to the Zn^2+^ or Zn^0^ [48]. Interstitial Zn atoms in ZnO were described in [49] as well as the transformation of Zn to Zn^+^ by capturing an electron. Moreover, Zn interstitials can diffuse [16]. Previously, blue PL has been observed in sputtered ZnO layers [50]. Blue PL in our samples seems to be related to the surface states. It disappeared after annealing in air.

Understanding the generation of photocurrents in semiconductors is essential for designing optoelectronic devices, particularly those affected by persistent photoconductivity. The process typically involves several stages: the absorption of incident (UV) light, the excitation of electron–hole pairs, and the transport of charge carriers. Excitation occurs when the energy of the incident light exceeds the semiconductor’s band gap, promoting electrons to the conduction band and generating electron–hole pairs, often influenced by impurities and defect states [51]. In our previous work [29], we observed high photocurrents under UV light in pristine ZnO films prepared by magnetron sputtering. However, the photocurrents in ZnO exhibited rapid attenuation, decreasing significantly during the first voltage scan and continuing to decline in subsequent measurements. This decay was initially attributed to either photo-corrosion or changes in the work function induced by oxygen vacancies; however, a definitive explanation has remained unclear. The Hall effect measurements presented in this paper suggest that our samples were n-type with a relatively low carrier concentration and low electron mobility. While low electron concentration could be useful for certain applications, the low mobility and persistent photoconductivity indicate potential issues with charge transport. In materials exhibiting persistent conductivity, charge carriers remain in an excited state long after the light source is removed [43].

## 5. Conclusions

The reactive pulsed mid-frequency sputtering with RF ECWR plasma enabled growth of highly optically transparent thin ZnO films on glass substrates. The bulk and surface PL has been distinguished by excitation using UV LEDs with wavelengths of 340 nm and 385 nm. The PL at low temperature (nominally 4 K) revealed several PL bands, including surface related blue PL in some samples. The red PL was presented in all samples, and it was linked to oxygen and zinc vacancy complexes, which could be controlled through annealing treatments. Hall effect measurements revealed n-type conductivity, though carrier mobility remained relatively low at about 1 cm^2^V^−1^s^−1^, probably due to the scattering of free carriers by high concentration of deep ionized defects. Persistent photoconductivity indicates the presence of defect-induced charge trapping mechanisms. Lattice disorder and deep defect concentration are more likely to occur in samples with higher surface roughness. These results suggest that the performance of optoelectronic devices needs to be improved. Future research should focus on refining the deposition process and post-deposition treatment to minimize the surface roughness and persistent photoconductivity, ensuring ZnO films meet the demands of advanced technological applications such as sensors.

## Figures and Tables

**Figure 1 nanomaterials-15-00590-f001:**
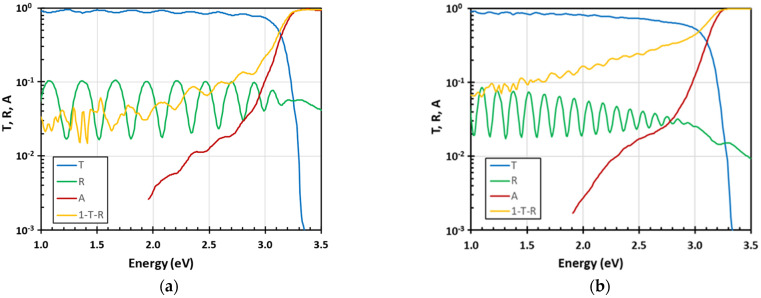
Optical transmittance (*T*), reflectance (*R*), and absorptance (*A*) spectra of selected thin films: (**a**) 1068 nm thick #3; (**b**) 2254 nm thick #4. For comparison, the 1-*T*-*R* spectra have been added.

**Figure 2 nanomaterials-15-00590-f002:**
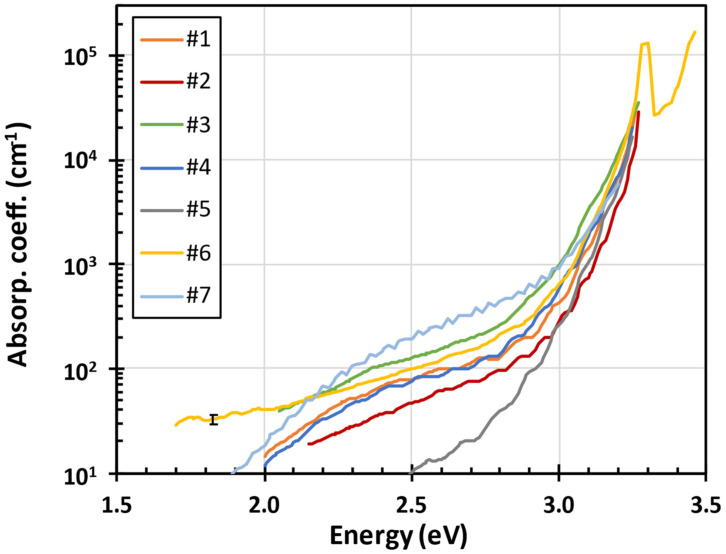
The optical absorption coefficient spectra of the ZnO layers listed in Table 1. The optical absorption coefficient below 10 cm^−1^ was too low to evaluate. Ab error bar is indicated on yellow curve at 1.8 eV.

**Figure 3 nanomaterials-15-00590-f003:**
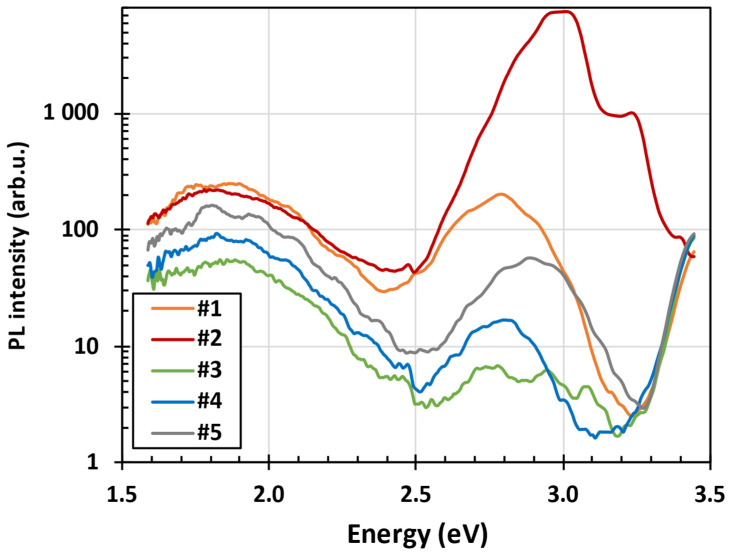
The steady-state PL spectra of the ZnO layers #1–#5, listed in Table 1, measured at low temperature (nominally 4 K) under UV LED340 illumination.

**Figure 4 nanomaterials-15-00590-f004:**
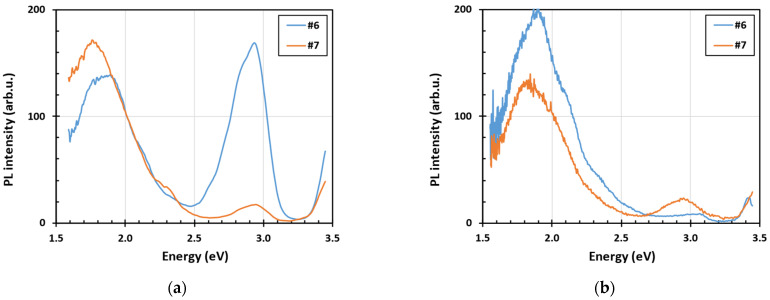
Steady-state PL spectra measured at low temperature (nominally 4 K) under the excitation wavelength 340 nm before (**a**) and after (**b**) 15 min annealing in air at 500 °C of two selected thin films: 1320 nm thick sample #6 and 4700 nm thick sample #7.

**Figure 5 nanomaterials-15-00590-f005:**
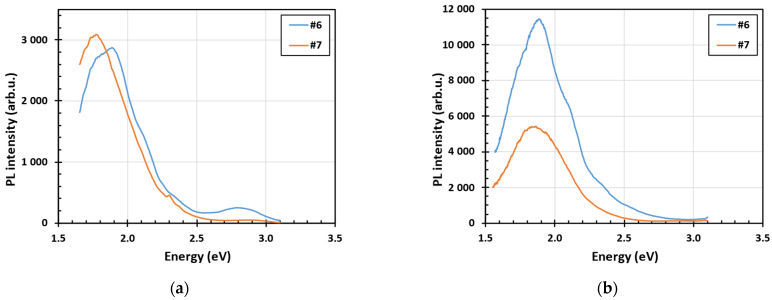
Stead-state PL spectra measured at low temperature (nominally 4 K) under the excitation wavelength 385 nm before (**a**) and after (**b**) 15 min annealing in air at 500 °C of two selected thin films: 1320 nm thick sample #6 and 4700 nm thick sample #7.

**Figure 6 nanomaterials-15-00590-f006:**
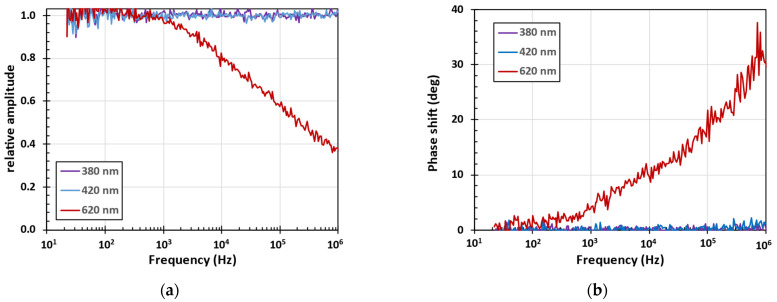
The frequency resolved PL of the sample #2 measured at room temperature by lock-in amplifier at selected emission wavelengths 380 nm (3.26 eV), 420 nm (2.95 eV) and 620 nm (2.0 eV) under LED360 excitation: (**a**) relative amplitude of the PL emission normalized on 1 at low frequencies; (**b**) relative phase shift between the excitation and emission signals.

**Table 1 nanomaterials-15-00590-t001:** List of thin film samples, substrates (soda lime glass SLG or fused silica glass FSG), and their basic characteristics: ECWR power *P*, layer thickness *d*, root-mean-square surface roughness *rms*, deposition rate, Urbach edge *E_U_*, optical absorption coefficient α at 2.5 eV and dark resistivity ρ.

Sample	*Subst.*	*P* (W)	*d* (nm)	*rms* (nm)	*Dep. Rate* (nm/min)	*E*_U_ (meV)	*α* (cm^−1^)	ρ (Ω.cm)
#1	SLG	0	1580	12	48	65	77	
#2	SLG	100	2316	50	100	58	48	
#3	SLG	200	1068	2	63	79	127	
#4	SLG	300	2254	15	138	72	77	
#5	SLG	380	2086	18	98	58	10	
#6	FSG	200	1320	160	44	64	100	4.6 × 10^5^
#7	FSG	200	4700	300	130	109	200	1.1 × 10^2^

## Data Availability

The data are available in a public ASEP repository of the Czech Academy of Sciences (https://doi.org/10.57680/asep.0618674) accessed on 3 April 2025.

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
