# Peer review of "Optical and Transport Properties of ZnO Thin Films Prepared by Reactive Pulsed Mid-Frequency Sputtering Combined with RF ECWR Plasma"

_nanomaterials, 2025, doi:10.3390/nano15080590_

Round 1
Reviewer 1 Report
Comments and Suggestions for Authors
In this paper, Remes et al. comprehensive study on the optical and transport properties of polycrystalline ZnO thin films prepared using reactive pulsed mid-frequency sputtering with RF electron cyclotron wave resonance (ECWR) plasma. They found red and blue emissions. So, I recommend its publication, with suggested changes, as below:
- Introduction section, authors need to specify the advantages of electron cyclotron wave resonance (ECWR) plasma assistance compared to other deposition techniques based on ZnO thin layers.
- Authors need to clarify the relation between ECWR power P(W) and layer thickness d(nm) in Table 1.
- Table 1, author mentioned root-mean-square roughness rms, how to find rms values.
- Please check Figure 4, the 1064 nm thick #3 different from Table 1.
Author Response
In this paper, Remes et al. comprehensive study on the optical and transport properties of polycrystalline ZnO thin films prepared using reactive pulsed mid-frequency sputtering with RF electron cyclotron wave resonance (ECWR) plasma. They found red and blue emissions. So, I recommend its publication, with suggested changes, as below:
- Introduction section, authors need to specify the advantages of electron cyclotron wave resonance (ECWR) plasma assistance compared to other deposition techniques based on ZnO thin layers.
The defects are usually a problem in reactive sputtering of ZnO films and their concentration is usually higher than in CVD methods. It is shown that additional ECWR can influence the defect concentration in a certain range, as can be seen from the luminescence results. Some values of ECWR power positively enhanced intensity of green luminescence as it can be seen in Fig. 3 for sample #2 done at Pecwr=100 W. We believe that higher concentration of oxygen atoms in case of applied ECWR plasma can compensate some defects by reactive nature of these O atoms. On the other hand, for higher ECWR power, higher plasma density may be responsible for creation of other defects. The plasma parameters should be properly tuned to obtain optimum conditions for semiconductor ZnO growth.
- Authors need to clarify the relation between ECWR power P(W) and layer thickness d(nm) in Table 1.
The deposition rate calculated from the film thickness and the deposition time has been added inro Table 1. - Table 1, author mentioned root-mean-square roughness rms, how to find rms values.
rms roughness values has been added into Table 1. They were evaluated from the suppression of the interference fringes in reflectance spectra in low-absorbing near infrared region using effective media approximation EMA model. To keep the table concise, the last two columns, mentioning the free electron concentration and the electron mobility of the sample #7 were removed and are mentioned only in the text. - Please check Figure 4, the 1064 nm thick #3 different from Table 1.
The thickness of the sample #4 is 1068 nm and it has been checked and corrected in all text. Thanks for reading the text carefully to report an error.
Reviewer 2 Report
Comments and Suggestions for Authors
This manuscript presents a study on the optical and transport properties of ZnO thin films fabricated using a reactive pulsed mid - frequency sputtering technique combined with RF ECWR plasma. The research is relevant to the field of optoelectronic materials, and the results could potentially contribute to the development of ZnO - based devices. However, several aspects need to be addressed to enhance the quality and clarity of the manuscript.
1. The connection between the previous work and the current study could be more explicitly stated. It is not entirely clear what specific gaps the authors aim to fill. Highlighting the limitations of previous studies more precisely would justify the need for this new investigation.
2. For optical transmittance, reflectance, and PDS measurements, it is not clear how the liquid immersion (Florinert FC72) might affect the measurement accuracy, especially considering potential refractive index mismatches. A discussion on the uncertainty introduced by this method would be valuable.
3. In the photoluminescence measurements, the calibration procedures for the different setups (steady - state PL and time - resolved PL) could be more detailed. How often was the calibration performed? Were there any quality control measures in place to ensure accurate calibration?
4. In the optical absorption coefficient spectra (Figure 2), it would be helpful to provide error bars to indicate the uncertainty in the measurements. This would give readers a better sense of the reliability of the data.
5. When presenting the Hall and photo - Hall effect measurements, more data points could be included, especially for the photocurrent under different illumination conditions. This would allow for a more comprehensive analysis of the persistent photoconductivity phenomenon.
6. In Table 1, while the correlation between Urbach energy and the optical absorption coefficient at 2.5 eV is presented, a more in - depth analysis of the other parameters (such as surface roughness, film thickness) in relation to the optical and transport properties would be interesting. Are there any trends or correlations that can be further explored?
7. The discussion on the origin of defects in ZnO films is somewhat scattered. A more systematic analysis of how different growth conditions (such as ECWR power, deposition temperature) and post - treatment (annealing) affect the types and concentrations of defects would be more informative. Additionally, the relationship between the observed defects and the electrical and optical properties should be more quantitatively explored.
8. Although some references are cited, the comparison of the current results with previous studies is not comprehensive enough. For example, when discussing the n - type conductivity and low carrier mobility, a more detailed comparison with similar studies on ZnO thin films prepared by different methods would help to better understand the significance of the current findings.
Author Response
This manuscript presents a study on the optical and transport properties of ZnO thin films fabricated using a reactive pulsed mid - frequency sputtering technique combined with RF ECWR plasma. The research is relevant to the field of optoelectronic materials, and the results could potentially contribute to the development of ZnO - based devices. However, several aspects need to be addressed to enhance the quality and clarity of the manuscript.
- The connection between the previous work and the current study could be more explicitly stated. It is not entirely clear what specific gaps the authors aim to fill. Highlighting the limitations of previous studies more precisely would justify the need for this new investigation.
We have completely rewritten the Introduction to more explicitly state the connection between the previous work and the current study. In particular, we have added at the end of the Introduction the statement “ Presented work deals with further study of parameters of ZnO thin films deposited by MF pulsed reactive magnetron sputtering under similar conditions as in our previous publications. Other parameters like Hall mobility, carrier concentration, absorption coefficient in low absorption region measured by PDS method and low temperature PL excited by UV LEDs were measured and discussed related to deposition. The bulk and surface PL has been distinguished by excitation wavelengths 340 (above) and 385 nm (below optical absorption edge.”
For optical transmittance, reflectance, and PDS measurements, it is not clear how the liquid immersion (Florinert FC72) might affect the measurement accuracy, especially considering potential refractive index mismatches. A discussion on the uncertainty introduced by this method would be valuable.
The index of refraction of Florinert FC72 is 1.26, whereas ZnO has the index of refraction about 2 in the near infrared region and higher in UV region. Therefore the index of refraction of Florinert FC72 plays a small role in our measurements. Moreover, the index of refraction of transparent liquid has been taken into account during evaluation of optical spectra using commercial FilmWizard software.
3. In the photoluminescence measurements, the calibration procedures for the different setups (steady - state PL and time - resolved PL) could be more detailed. How often was the calibration performed? Were there any quality control measures in place to ensure accurate calibration?
Both PL setups were spectrally calibrated with a #63358 halogen lamp (Oriel Instruments, subsidiary of Newport Corporation, Stratford, CT, USA). The additional correction [23] of the steady-state setup was related to the Jacobian conversion of wavelength and energy scales for quantitative analysis of emission spectra. This statement has been added into text for clarity. The spectral calibration of the time-resolved setup plays no significant role, because this setup was used for time resolved measurements at selected wavelengths only.
4. In the optical absorption coefficient spectra (Figure 2), it would be helpful to provide error bars to indicate the uncertainty in the measurements. This would give readers a better sense of the reliability of the data.
Unlike optical absorptance A calculated at as 1-T-R from transmittance T and reflectance R spectra, PDS measures A directly by measuring temperature on the sample surface due to optical absorption. It is therefore much less sensitive to the optical scattering and the optical absorption is detectable down to 10-4. corresponding to the optical absorption coefficient 10 cm-1 for about 1000 nm thick film on glass substrate. for about 1000 nm thick thin films on glass substrates. The optical absorption coefficient was calculated from PDS spectra using previously calculated index of refraction and thin film thickness. Errorbars of the absolute value of the optical absorption coefficient were estimated from the residual interference fringes observable in the optical absorption coefficient spectra shown in Figure 2. This explanation is now included in the revised manuscript.
When presenting the Hall and photo - Hall effect measurements, more data points could be included, especially for the photocurrent under different illumination conditions. This would allow for a more comprehensive analysis of the persistent photoconductivity phenomenon.
In our research, we observed that after switching on the illumination, photoconductance reached a maximum and then decreased slightly, while it did not reach a completely stable value even within several tens of minutes. This caused the photo-Hall signal, measured against the background of drifting photocurrent, to be rather noisy. This observation was also consistent with the behavior of the dark Hall measurement, where a stable Hall signal was obtained only after several hours from inserting the sample into the measurement chamber.
This circumstance, together with not fully optimized homogeneity of the light beam, prevented us from performing extensive photo-Hall measurement as it would not be sufficiently reliable for the detailed analysis of the persistent conductivity phenomenon.
We plan further measurements devoted to the time dependence of conductivity and Hall effect in ZnO prepared by the described method using different illumination and other environmental conditions, which could help us to improve the measurement settings for detailed investigation of the photo-Hall effect in low-mobility samples.
In Table 1, while the correlation between Urbach energy and the optical absorption coefficient at 2.5 eV is presented, a more in - depth analysis of the other parameters (such as surface roughness, film thickness) in relation to the optical and transport properties would be interesting. Are there any trends or correlations that can be further explored?
The Table 1 and the corresponding part of Conclusions was rewritten.
The high correlation coefficient R = 0.93 between the Urbach edge slope EU and the optical absorption coefficient a(2.5 eV) indicates the high degree of correlation between valence band disorder and the concentration of deep defect states. The significant correlation has been found also between sample thickness and rms roughness (R = 0. 76), a(2.5 eV) and rms roughness (R = 0. 74), between EU and rms roughness (R = 0. 73) and between EU sample thickness and (R = 0. 74). The Urbach energy extracted from the absorption spectra increases with increasing lattice disorder, indicating stronger tailing of the density of states into the band gap. The optical absorption coefficient at energy 2.5 eV, well below the optical absorption edge at 3.3 eV, is related to deep electron states in the energy gap acting as recombination centers. It appears that lattice disorder and deep defect concentration are more likely to occur in samples with higher surface roughness.
- The discussion on the origin of defects in ZnO films is somewhat scattered. A more systematic analysis of how different growth conditions (such as ECWR power, deposition temperature) and post - treatment (annealing) affect the types and concentrations of defects would be more informative. Additionally, the relationship between the observed defects and the electrical and optical properties should be more quantitatively explored.
The defects are usually a problem in reactive sputtering of ZnO films and their concentration is usually higher than in CVD methods. It is shown that additional ECWR can influence the defect concentration in a certain range, as can be seen from the luminescence results. Some values of ECWR power positively enhanced intensity of green luminescence as it can be seen in Fig. 3 for sample #2 done at Pecwr=100 W. We believe that higher concentration of oxygen atoms in case of applied ECWR plasma can compensate for some defects by reactive nature of these O atoms. On the other hand, for higher ECWR power, higher plasma density may be responsible for creation of other defects. The plasma parameters should be properly tuned to obtain optimum conditions for semiconductor ZnO growth.
There was no correlation observed between the Urbach energy and surface roughness, which is now included in the text. The surface roughness values. has been included into Table 1. The root-mean-square (rms) of surface roughness was evaluated from the suppression of the interference fringes in the reflectance spectra in the near infrared region using the Effective media approximation (EMA) mode. It was empirically shown that for the rough surfaces with the high spatial frequencies the rms is approximately halve of EMA thickness , which approximately coincides with the result achieved theoretically for the one-dimensional rough surface at normal incidence of light . This explanation is now included in the revised manuscript.
- Although some references are cited, the comparison of the current results with previous studies is not comprehensive enough. For example, when discussing the n - type conductivity and low carrier mobility, a more detailed comparison with similar studies on ZnO thin films prepared by different methods would help to better understand the significance of the current findings.
New references were added to the Introduction, Results and Discussion. The revised paper includes 51 citations instead of 37 in the original version.
Round 2
Reviewer 2 Report
Comments and Suggestions for Authors
The author has addressed all the reviewer's comments and it is recommended to be published in Nanomaterials.